# Qishta—A Lebanese Heat Concentrated Dairy Product Characteristics and Production Procedures

**DOI:** 10.3390/foods9020125

**Published:** 2020-01-24

**Authors:** Mustapha Najib, Mohamad Walid Hallab, Karim Hallab, Zaher Hallab, Guillaume Delaplace, Monzer Hamze, Nour-Eddine Chihib

**Affiliations:** 1Health and Environment Microbiology Laboratory, Doctoral School of Sciences and Technology, Faculty of Public Health, Lebanese University, Tripoli, Lebanon; mustapha.najib@inra.fr (M.N.); mhamze@monzerhamze.com (M.H.); 2UMET CNRS Laboratory, INRAE, UMR 8207-UMET-PIHM, Lille University, 59652 Villeneuve d’Ascq, France; guillaume.delaplace@inra.fr; 3HALLAB 1881 s.a.l, Tripoli, Lebanon; walid@hallab.com.lb (M.W.H.); Karim@hallab.com.lb (K.H.); Zaher@hallab.com.lb (Z.H.)

**Keywords:** Qishta, heat treatment, milk, casein, whey protein

## Abstract

This study aims at exploring the chemical composition of a traditional Lebanese dairy product known as Qishta, describing the process of how to prepare it and understanding the mechanisms leading to its formation. The process of making Qishta can be divided into two phases: a hot phase during which milk is heated in a stainless-steel large shallow vessel, and a cold phase consisting of draining, cooling and packaging. According to milk temperature, two reaction zones were identified: zone A with an average temperature of 100 °C, and zone B with an average temperature of 60 °C. The results showed that Qishta had a moisture, fat, protein, lactose and ash content of 68%, 11.7%, 12.1%, 5.4% and 1.6%, respectively. Our findings showed that Qishta is a lipoprotein product having an equal amount of fat and proteins (≈12%); this composition is almost similar to that of Ricotta cheese made from whole milk. In addition, our results assert that the interactions between caseins and whey proteins lead to gel formation. Milk initial fat percentage had a significant effect on Qishta production. The highest yields were obtained when the initial fat percentage was 3.6% (182.5 g of Qishta).

## 1. Introduction

During the last decade, Lebanese dairy manufacturing occupied an important role in the agro-industrial field. The local production of milk and dairy products covers only a quarter of the need [1]. Hallab 1881 company, one of the oldest family enterprises located in the north of Lebanon, is known for their sweets and dairy products such as Qishta. Hallab company is the pioneer of oriental sweets in Lebanon, producing more than 200 t of Qishta per year. Although the historical origin of this unique product is not really known, it began as a homemade product. In the Middle Eastern region, different spellings are used for Qishta such as Kishta, Kashta or Ghishta. In our study, the spelling Qishta will be used according to Kassaify et al. [2]. During the Othman empire, the product spread in the region and similar products existed in Turkey (Kaymak) [3] and in Serbia (Kajmak) [4].

Within the same context, Qishta is a popular Middle Eastern dairy product prepared using traditional heating and skimming processes [5]. The lack of milk collection industries in Lebanon has led the producers of Qishta to use milk powder as the raw material instead of fresh milk. Consequently, milk powders are imported from France, Denmark, the Netherlands, and Eastern Europe (particularly from the Czech Republic) [1]. The process of making Qishta can be divided into two phases: (i) the hot phase and (ii) the cold phase (Figure 1). The hot phase consists of heating acidified milk for two to three hours (performed by pouring and boiling acidified milk into open shallow vessels). Approximately 10 min after the beginning of heating, a skin layer (Figure 2a) appears on the milk surface (Qishta skin) as a result of the combined influence of: (i) protein denaturation and fat coalescence on the milk surface leading to a concentrated layer of both components; and (ii) intense evaporation of water from the surface due to the increase in milk temperature. The skin formed will be broken, creating a pathway for the aggregates (second layer) formed in the heated zone and then gathered at the opposite side of the flame (Figure 2b). These aggregates will be gathered with the skin and drained in order to form Qishta (Figure 2c). The cold phase includes draining, cooling and Qishta packaging. The product has a chemical composition and texture comparable to those of Ricotta cheese prepared from whole milk, with a slightly higher amount of fat and protein. In Lebanon, the process of making Qishta varies according to the region, however the final product is almost the same. The process described in this article is in accordance with that reported by Kassaify et al. [2].

Qishta can be related to cheese family (especially to cream cheese varieties due to the presence of milk proteins and their particular coagulation during the Qishta making process). Qishta is usually not consumed fresh; however, it is further processed in order to prepare large oriental sweet varieties such as Knefe, Mafrouke and so forth. Knefe is made by filling Qishta between 2 layers of roasted semolina, poured with cane syrup and sprinkled with ground pistachio (Figure 3a), while Mafrouke is a mixture of roasted semolina dough and cane syrup topped with Qishta (Figure 3b) and fried nuts. Overall, Hallab 1881 company offers more than 50 products that contain Qishta.

Despite the high consumption of Qishta in Lebanon, only one article has described the microbiological and chemical profile of this product [2]. The present work aims to determine the chemical composition of Qishta, to monitor the temperature profile of milk during the heating process and to understand the mechanism of Qishta formation.

## 2. Materials and Methods

### 2.1. Materials

Milk was provided by Lactel (Laval, France). It is an ultra-heat treated (UHT) whole milk with the following composition: 3.6% fat, 3.2% protein, 4.8% lactose and 1% ash. Lactic acid was purchased from Fischer Scientific (Loughborough, UK). The plate (stainless-steel, 316 L) used in Qishta preparation was provided by Nafco (Baouchriyeh, Lebanon) and the burner was provided by Brûleur AEM (Chelles, France) equipped with a pressure regulator.

### 2.2. Qishta Preparation Procedure

The plate (shallow vessel) used in Lebanon for Qishta production has a diameter of 1 m, a capacity of 9 L and a thickness of 2 mm. A new plate, with a 0.5 m diameter, 3 L capacity and a 2 mm thickness was adapted during our study by Nafco, in order to produce Qishta at a smaller scale. Lactic acid (1.5 mL) was added to 4 L of UHT milk in order to decrease the pH from 6.7 to 6.4 (the same as the procedure used at Hallab 1881 company). The traditional preparation process consists of heating milk for 2 h and simmering the aggregates formed at the surface. During the heating process and depending on the evaporation rate, milk is added in order to readjust its level in the plate. In our experiments, milk was not added in order to keep the milk’s composition well known, and therefore the process was interrupted after 25 min of heating. The remaining milk at the end of process was collected for further analysis. In Lebanon, this milk known as red milk and is revalorized into traditional dairy desserts.

### 2.3. Temperature Distribution

During heat treatment, the temperature distribution profile was monitored using 5 probes provided by ATC Mesures (Tourcoing, France). During the Qishta making process, the probes were immerged in the milk and their positions are indicated in Figure 4.

### 2.4. Physicochemical Analysis for Qishta

Qishta and red milk (residual milk at the end of the production according to the nomination used in Lebanon) were analyzed in triplicate for their compositions. Total nitrogen (expressed as protein equivalents) was quantified by the Kjeldahl method (International Dairy Federation, 2002). Ash content was determined by standard 27/1964 IDF, dry matter (DM) by adapting the oven drying method at 102 ± 2 °C (International Dairy Federation, 1982), and fat by applying the butirometric method (International Dairy Federation, 1986). Lactose content was determined using 3,5-dinitrosalicylic acid (DNS) [6]. Evaporation rate was monitored using a scale located under the plate during the heating process. The remaining milk in the plate was weighed every 2 min; therefore, the amount of water evaporated was calculated.

### 2.5. Static Light Scattering

The granulometric distribution was determined by laser light scattering using a MasterSizer 2000 (Malvern Instruments, Malvern, Worcestershire, UK) equipped with a 5 mW He–Ne laser operating at a wavelength of 633 nm. Samples were diluted into the Malvern cell (volume: 100 mL) with distilled water to reach appropriate obscuration (25%). Refractive indexes for solvent, particle and adsorption were 1.33, 1.3 and 0.1, respectively. These indexes were recommended by the manufacturer (Malvern).

### 2.6. SDS-PAGE Analysis

Samples of UHT milk (Lactel), red milk and Qishta were analyzed by SDS-PAGE electrophoresis under reducing (SDS-R) and non-reducing (SDSNR) conditions using the method described by Anema [7]. The resolving and stacking gel contained 15%, 12% and 4% acrylamide, respectively. Electrophoresis was performed using a vertical electrophoretic unit type TV200YK twin-plate, associated with the source voltage EV202. SDS-PAGE was executed at 30 V until the samples had completely left the stacking gel, then the voltage was increased to 90 V until the tracking dye reached 80% of the gel. Staining of the gel was performed in a 0.23% solution of Coomassie Blue R-250, containing 3.9% (*w*/*v*) Trichloroacetic acid, 6% (*v*/*v*) acetic acid and 17% (*v*/*v*) methanol for 90 min.

### 2.7. Milk Fat Effect

Bottled UHT skimmed milk (Lactel, Laval, France) was used in order to test the feasibility of producing Qishta without fat. Five milk samples with five different fat concentrations (3.6%, 2.6%, 1.6%, 0.6%, 0%) were prepared by mixing whole milk (3.6% of fat) with skimmed milk in order to test the effect of fat concentration on Qishta yield.

### 2.8. Statistical Analysis

One-way analysis of variance (ANOVA) was conducted using the Statistical Package for Social Sciences (SPSS) software for Windows (version 13.0, SPSS). A Duncan test was carried out to assess any significant differences between the means. Differences were considered statistically significant when *p* ≤ 0.05.

## 3. Results and Discussion

### 3.1. Chemical Composition of Qishta

The results obtained in Table 1 show the chemical composition of Qishta, UHT and red milk. Qishta has the following composition: moisture (68 ± 2%), fat (11.7 ± 0.6%), protein (12.1 ± 0.7%), lactose (5.4 ± 0.2%) and ash (1.6 ± 0.2%). These results were no different from those obtained in Lebanon (Hallab 1881 company). The concentration of protein and fat were much higher in Qishta than in Lactel milk, indicating a migration from milk to the gel during heat treatment. At the end of the process, the remaining milk was rich in protein and fat; however, this milk cannot be compared to that obtained in Lebanon, since the process was stopped after 25 min of heating.

Our findings asserted that Qishta is a partially dehydrated product with a composition similar to that of Ricotta cheese prepared from whole milk, with almost the same amount of fat and protein (Table 1) [8]. Our results were in accordance with those reported by Kassaify et al., [2], who found almost the same composition. In fact, the composition of the final product would vary according to the milk composition, hydration process, heating time, amount of milk added and intensity of heating. The high value of the solid content in the remaining milk can be explained by the high evaporation rate taking place during the heat treatment of milk (Table 1). During Qishta preparation, the concentration of milk increased proportionally with water evaporation. This latter is the main reason behind the addition of milk during the process; a low level of milk in the plate leads to an undesired burnt taste in the final product.

### 3.2. Temperature Distribution Profile

Temperature distribution was investigated during 25 min of heating. Figure 5 shows the milk temperature distribution according to the different positions of probes on the plate. The general trend of the temperature profile monitored by the five probes show some similarities. At first glance, two behaviors were observed according to the position of the probes, resulting in two reaction zones: A and B. In zone A, which is closer than zone B to the flame and where Qishta is produced, the temperature increased by 25 °C/min. Two minutes later, the rate became 3.24 °C/min and reached a stable temperature of 100 °C after 15 min of heating. In zone B, the kinetics of the temperature increase were slower than that obtained in the first zone with 2.9 °C/min in the first 3 min of heating. The temperature became stable at around 65 °C after 15 min of heating. The heterogeneity in the temperature distribution can be explained by the position of the burner on the edge of the plate and by the plate’s nature (stainless steel).

It has been reported that heat treatment induces many detrimental effects in emulsions [9,10]. During Qishta production, the heat treatment leads to fat coalescence, protein denaturation and finally to gel formation. Protein aggregation is considered as a key point during the process of making Qishta. Considering their solid primary chains, caseins are known to be heat resistant; they can withstand an intense heating of 140 °C for 15–20 min. When heated above 100 °C, the size of casein micelles decreases because of both kappa-casein (κ-casein) detachment from the micelle surface and colloidal phosphate liberation [11]. These phenomena probably took place in zone A.

Due to their globular structure, whey proteins (WP) are more sensitive to heat than caseins. Above 60 °C, they lose their tertiary structure and denature. This new unfolded structure exposes the amino acid groups and allows them to interact with other proteins through disulfide bonds [12,13]. In Qishta production, this is likely to happen in both zones A and B. Increasing the WP concentration leads to an enhancement of fat droplet coalescence as well as emulsion viscosity until the critical concentration of gelation is reached. This was later estimated at around 3 wt% of non-adsorbed WP [9]. Above 3%, the non-adsorbed denatured WP will connect the different fat droplets in a continuous network. Euston et al. [12] compared WP to a glue that connects the fat droplets in the continuous phase. However, below the critical concentration, WP are not able to play the glue role and therefore the gelation does not occur.

The aggregates or the gel formed on top of the milk contain fat, casein, lactose and WP. During heat treatment, β-lactoglobulin (β-lg) binds to κ-casein on the surface of casein micelles through disulfide bonds and hydrophobic interactions [14,15]. In fact, due to the severe temperature applied, β-lg dissociates and loses its tertiary and part of its secondary structure, leaving the free thiol group (Cys121) exposed to the interactions with κ-casein through disulfide bridges [16].

The intensity and the period of heating have a significant impact on the process undergone by the milk. The process of making Qishta induced an increase in milk concentration due to water evaporation. Figure 6 shows that the evaporation rate followed an exponential increase estimated at 0.6 mL/min. After 20 min of heating, around 400 mL of water (≈13%) had evaporated, leading to an increase in milk concentration from 10% to 14%.

### 3.3. Emulsion Stability to Heat Treatment

#### Static Light Scattering (SLS)

Figure 7 shows the changes in particle size distribution of milk samples taken from the heated zone induced by the thermal treatment. Before heating, the average size of the particles used in this study was about 1.1 µm. When the milk was heated, the particle size distribution changed: larger particles (mostly aggregates) were observed after 5 min, with 5% of the total particles having a diameter between 258 µm and 750 µm. The volume weighted particle size distributions (D90) at this moment was 0.63 µm (90% of the particles had a diameter less than 0.63 µm). The increase in the droplet size could be explained by the fat coalescence phenomena or by the interactions that occur between fat globules and proteins during the heat treatment, or by a combination of both phenomena. Raikos [17] has noticed an increase in the particle size to 1−10 µm when milk was heated at 140 °C for 80 s. This increase was attributed to the interactions that occurred between non-adsorbed protein molecules in the serum phase and proteins adsorbed at the interface of fat globules. A trimodal distribution was observed after 10 min of heating with a continuous increase of the particle size where (D90) reached 1.94 µm. The heterogeneous temperature distribution and the convection forces existing throughout the plate led to the formation of different aggregate sizes represented by the trimodal distribution.

Dickinson [18] defined emulsion stability by its resistance to change over time. Our results indicated that protein aggregation took place when the milk was heated. This protein aggregation might be involved with fat participation in Qishta formation. In milk emulsions, the composition and concentration of proteins present in the continuous phase have a primordial effect on the protein adsorption at the fat membrane. In fact, during heat treatment, caseins and WP are in competition to be adsorbed on the surface of fat droplets [19,20]. The particle size distribution is an important feature of many products, ranging from powder suspensions to emulsions, determining not only the physical properties such as flowability, but also the visual aspect and sensorial properties [21].

### 3.4. Identification of Major Proteins in Qishta

The electrophoresis analysis of UHT milk, Qishta and the remaining milk (red milk) was performed under reducing and non-reducing conditions (Figure 8). Under non-reducing conditions, the results showed that the major whey protein β-lg was absent in the studied samples. Similar protein patterns of high molecular weight (HMW) fractions (50 kDa to 150 kDa) were identified in all samples under both different conditions. However, the intensity of these bonds was higher in non-reducing conditions. Between 50 kDa and 75 kDa, one of these patterns could be attributed to bovine serum albumin BSA (66 kDa). Concerning the UHT milk, the high molecular fraction of 150 kDa, which appeared under non-reducing conditions, was almost absent under reducing ones. Nevertheless, it can be clearly observed that the stacking gel obtained under the non-reducing condition contained some protein fractions. These fractions, with a very high molecular weight, were lost for analyses. Under reducing conditions, the most relevant fraction was that attributed to β-lg, with a much higher intensity than in the non-reducing gel; thus, the intensity of the casein patterns had also increased. Concerning the HMW patterns, the intensity obtained under reducing conditions decreased differently in each of the three samples. In fact, the reducing agent (β-mercaptoethanol) was not able to break all links between the aggregates which probably indicates the presence of new bonds, other than disulfide bridges between proteins. The analysis of the composition of the HMW patterns has revealed the presence of lysinoalanine and lanthionine crosslinks (data not shown). Results showed that the heating process during Qishta production induced the formation of HMW proteins as a result of the interactions between WP and caseins via disulfide bridges, as reported elsewhere [14,15].

### 3.5. Effect of Temperature on Milk Fats

Milk fat content influences the physicochemical, sensorial and quality characteristics of Qishta. Milk heat treatment induces numerous changes in the milk fat globule membrane (MFGM), whose role is to protect the fat globule from coalescence, denaturation and interactions with serum proteins via sulfhydryl–disulfide interchange reactions [22]. Figure 9 shows a supposed model of the interaction mechanisms that occur during the heat treatment of milk.

Upon heat treatment, β-lg and α-lactalbumin (α-la) bind to the fat globule [23,24]. However, the mechanism by which these proteins interact with the fat globule is still not evident.

When milk is heated above 70 °C, denaturation of MFGM proteins and exposure of various amino acid residues, particularly cysteine, take place. Thus, H_2_S is released (which results in the development of an off-flavor) and disulfide interchange reactions occur with WP. At high temperature (>100 °C), this will lead to the formation of a denatured WP layer that will adsorb on the MFGM. This adsorbed layer, in the presence of lactose, will participate in a Maillard reaction, while the cysteine could be involved in dehydroalanine formation by β-elimination and can react with lysine or cysteine and form lysinoalanine and lanthionine, respectively [25].

The extent heat treatment applied during the process of Qishta preparation results in the destruction of the MFGM, therefore increasing the susceptibility of fat to oxidation.

Houlihan et al. [26] stated that with the increase in the heating time (from 2.5–20 min at 80 °C) the amounts of β-Lg and α-La adsorbed on the MFGM increased, however the amounts of phospholipids and triacylglycerols decreased. In addition, a small amount of κ-casein was also present, and this component increased during heating. These results indicate that MFGM is involved in heat-induced interactions with milk proteins, in particular β-Lg and κ-casein, and that the amount of these components that adsorb onto the membrane depends on the extent of the heat treatments. κ-casein can either interact directly with MFGM components, or with β-Lg through disulfide interchange during heating.

### 3.6. Effect of Fat on Qishta Formation

#### Skim Milk Experiment

Lactic acid (1.5 mL) was added to skimmed milk in order to decrease its pH. Ten minutes after heating, the milk started to burn. There was no aggregate formation except a skin at the milk surface. Figure 10 shows that the milk initial fat percentage has a significant effect on Qishta production. The highest yield was obtained when the initial fat percentage was 3.6% (182.5 g). However, when the initial fat percentage was 1.6%, Qishta yield decreased from 182.5 g to 171 g. Figure 10 also shows that when the fat percentage increased in milk from 0.6% to 1.6%, the yield increased from 118 g to 134 g. A significant difference (*p* ≤ 0.05) was observed between the yields, indicating that the milk fat concentration has a significant impact on the yield. Increasing the fat percentage from 0.6% to 3.6% has been shown to have a positive effect on the production yield of Qishta.

## 4. Conclusions

Qishta is a specific dairy product with a non-consistent composition that varies depending on handlers, raw material, and the process applied to its production. Different Lebanese dairy industries have not succeed to replace the traditional process by an advanced one in order to resolve the problem of the consistency and the limited quantity produced. Thus, in order to improve the process of the production of Qishta, the composition and the mechanism must be well studied. Despite the beliefs that Qishta contains a high fat content, we demonstrated that it actually holds an equal amount of fat and protein and a texture closer to cheese rather than cream products. Increasing the milk fat concentration has proven to have a positive impact on the yield. Further analysis could be focused on the impact of increasing the fat concentration above 3.6%. This study must be coupled with an analysis of the chemical composition and the organoleptic properties of Qishta.

Further research could also be focused on investigating the structural differences between the two Qishta layers and the interactions between the proteins and fat forming the gel network.

## Figures and Tables

**Figure 1 foods-09-00125-f001:**
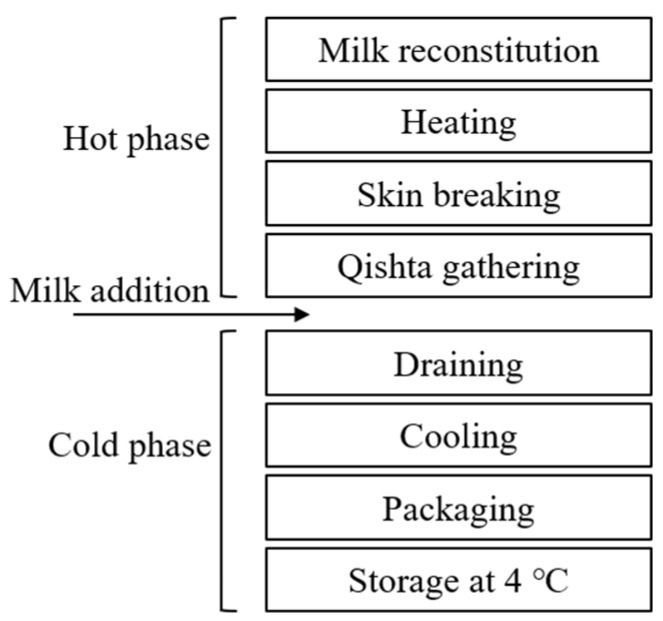
Key steps of Qishta process as observed at Hallab 1881 s.a.l. (Limited Anonymous Society).

**Figure 2 foods-09-00125-f002:**
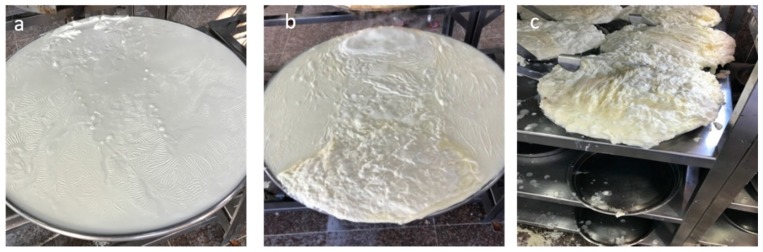
(**a**) Skin formation (**b**) Aggregates gathering (**c**) Qishta drainage.

**Figure 3 foods-09-00125-f003:**
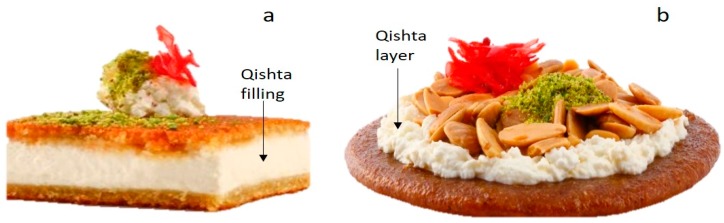
(**a**) Knefe, (**b**) Mafrouke prepared at Hallab 1881 company.

**Figure 4 foods-09-00125-f004:**
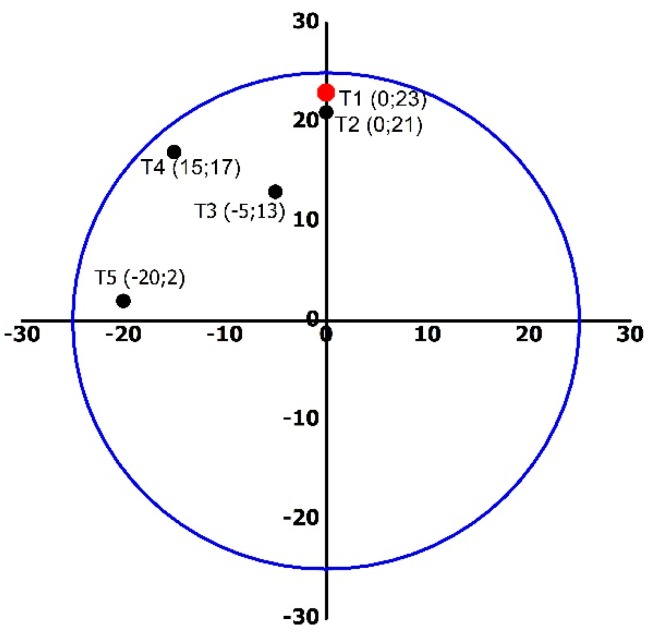
Position of the probes in the plate (cm). Probes are indicated by the letter T and are positioned in the plate according to the axes X and Y drawn on the plate. T1 is positioned above the burner. T2, T3, T4 and T5 have the following coordinates respectively (0; 21), (−5; 13), (−15; 17) and (−20; −2).

**Figure 5 foods-09-00125-f005:**
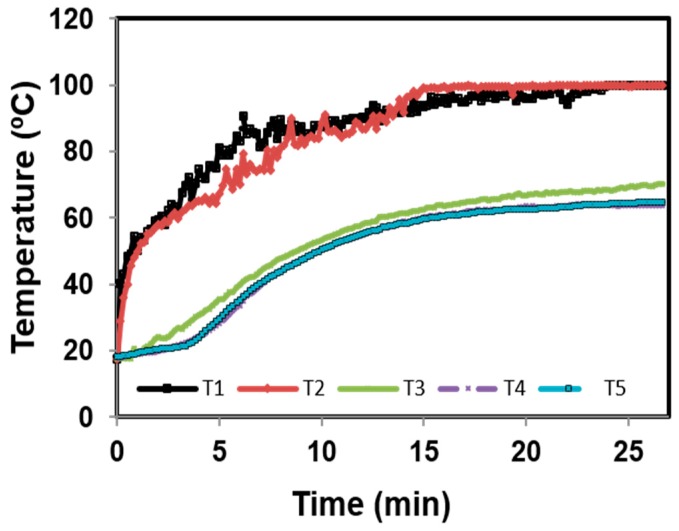
Temperature distribution profile during the Qishta formation: T1: blue circle, T2: red square, T3: green triangle, T4: purple cross, T5: yellow diamond.

**Figure 6 foods-09-00125-f006:**
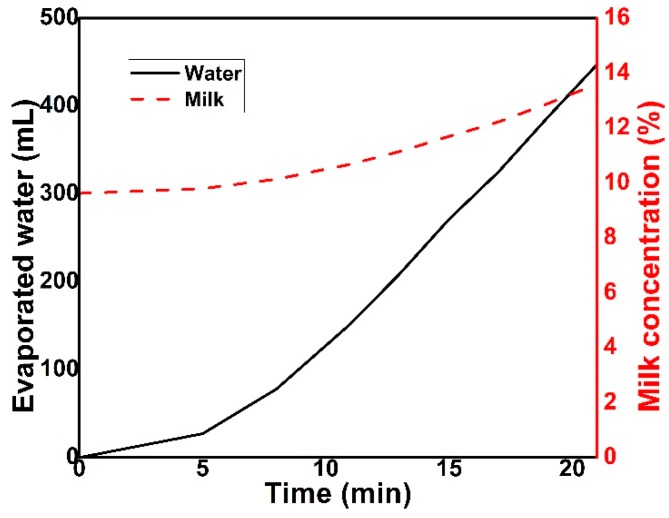
Relationship between the water evaporated and the concentration of milk during the process of Qishta production. Measurements were made in triplicate.

**Figure 7 foods-09-00125-f007:**
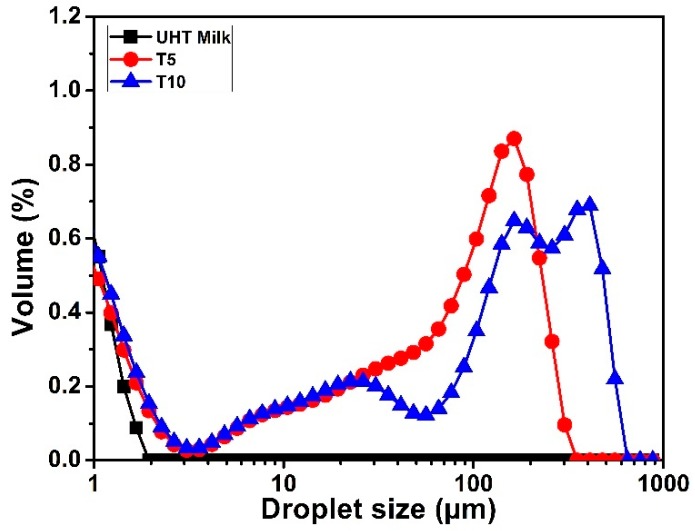
Follow-up of the particle size distribution using static light scattering (SLS) analysis during the first 10 minutes of heating. UHT milk before heating (control) ■ after 5 min of heating ● and after 10 min of heating **▲**.

**Figure 8 foods-09-00125-f008:**
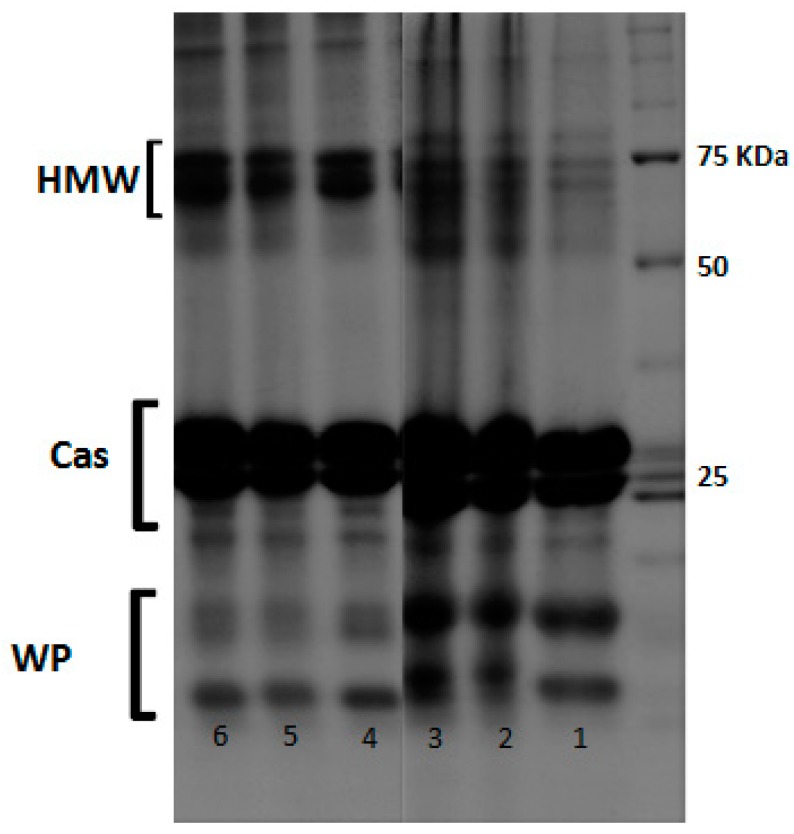
SDS electropherogram under reducing (1−3) and non-reducing conditions (4−6). 1,4: UHT milk; 2,5: red milk; 3,6: Qishta. Cas: casein, WP: whey protein, HMW: high molecular weight.

**Figure 9 foods-09-00125-f009:**
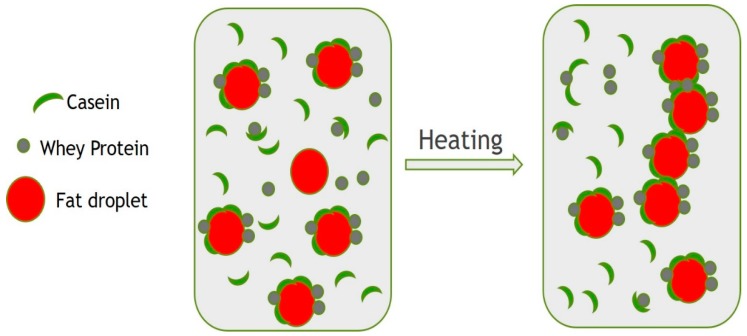
Interactions between different milk components during heat treatment.

**Figure 10 foods-09-00125-f010:**
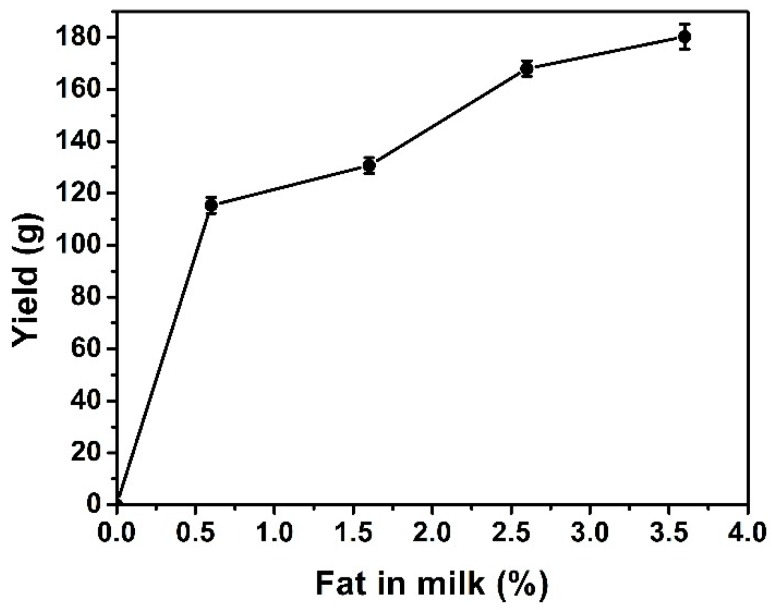
Yield average of Qishta as determined depending on the initial fat concentration in milk (0.6%, 1.6%, 2.6%, and 3.6%). Measurements were made in triplicate.

**Table 1 foods-09-00125-t001:** Chemical composition of raw milk, Qishta and red milk.

Sample	Moisture%	Fat%	FDM% ^1^	Protein%	PDM% ^2^	Lactose%	Ash%
Raw Milk	87.4	3.6	31	3.2	27.5	4.8	1
Qishta	68 ± 2	11.7 ± 0.6	36.6 ± 2.1	12.1 ± 0.7	37.2 ± 1.5	5.4 ± 0.2	1.6 ± 0.2
Red Milk	82.0 ± 0.1	4.2 ± 0.1	21 ± 1	4.0 ± 0.2	20.0 ± 1.7	7.0 ± 0.2	0.8 ± 0.1

^1^ FDM: fat in dry matter, ^2^ PDM: protein in dry matter.

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
