# Peer review of "Qishta—A Lebanese Heat Concentrated Dairy Product Characteristics and Production Procedures"

_foods, 2020, doi:10.3390/foods9020125_

Round 1

Reviewer 1 Report

The paper describes the preparation and chemical composition of the Middle Eastern dairy product Qishta. It contains some new information over that published in other papers such as the cited paper by Kassaify et al. It contains reasonable descriptions of the process and characteristics but provides limited evidence for the proposed mechanisms of interactions, for example fat-protein interactions.

Some specific comments are:

Title. The word “dessicated” means having had all moisture removed, i.e., dried out, and is therefore not appropriate for a product with 67.5% moisture. I suggest “concentrated” would be better

Line 18. I suggest adding “made from whole milk” after “ricotta cheese” as ricotta can also be made from whey, in which case its composition is very different

Line 41. Remove “freshly”

Line 50 states that the milk used was UHT milk but all through the paper the term “raw milk” is used. I believe the term “UHT milk” should be used throughout as the electrophoretogram for “raw milk “ in Figure 6 looks more like what one would expect for UHT milk, not raw milk. Authors to make corrections throughout the paper

Section 2.2. There needs to be mention of draining the residual milk in this section

Line 100. Was the “bottled skimmed milk” pasteurised? I presume it was not UHT treated as was the milk for the rest of the trials.

Line 110. Fat does not undergo “denaturation”

Section 3.1 should be under “Materials and Methods” or “Introduction”. It does not contain “Results” of this research.

Section 3.1 the “two Qishta layers” mentioned in Conclusions need to be introduced here

Figure 2. I suggest this figure be split into two figures, the left-hand side in one and the right-hand side in the other. The reason for this is that there is no relationship between the two sides.

Figure 2. I can see no value in including photo “d” as “Milk addition for a second production” has not been mentioned in the method used for production of the product in this research

Line 131. I do not agree with “lipoprotein” as a description of the product. Lipoprotein usually refers to material with only lipids and proteins; Qishta contains lactose and minerals in addition to lipids and proteins

Line 137. I suggest replacing “intense” with “high”

Line 173. Lactose should be added here

Line 183. In my copy this line is obscured by the figure

Line 190. The authors should check the 0.1 μm figure. This would apply to a skim milk as it is the size of casein micelles but the milk used contained fat globules which in (homogenised) UHT milk average around 1 μm

Line 195. What is the explanation for the trimodal distribution?

Line 223. Indicate what “new bonds” might be formed. There is a lot of literature on them.

Lines 228-229. The caption is confusing. Please adjust as follows: change “1-4 raw milk 2-

5 red milk 3-6 Qishta” to “1, 4 raw milk 2, 5 red milk 3, 6 Qishta”

Lines 241-242. The authors need to provide evidence of this?

Line 243. What evidence is there for this occurring in this system? What “other unfolded monomers” are the authors referring to? Note that reference 25 refers to UHT heating conditions which are very different from those for Qishta where the temperature does not exceed 100°C.

Lines 255-256 include the words “indicating that the optimal fat percentage in milk for Qishta production was between 2.6 and 3.6%” This statement is not valid as 3.6% was the highest percentage fat used. Based on Figure 8, the optimum yield would be expected to be at fat percentages >3.6%, not between 2,6 and 3.6%

Lines 264-267. Delete these lines

References. Please edit references to: replace non-family names with initials; Remove unnecessary upper-case letters from titles of papers; remove price of book at reference 18; correct format of references 11, 16 and 21; remove 28 as a reference.

Reviewer 2 Report

In the attachment

Reviewer 3 Report

- Line 43: The authors may want to add more background information about QIshta, for those readers who are not familiar with the product.

- Line 50: Why did the authors choose to use UHT milk when it is stated in the introduction that Qishta is usually made from milk powder?

- Line 103: Was any statistical analysis completed on the results?

- Line 265: There is no discussion of the results.

- Line 269: Conclusion: which stakeholders or industry would be interested in the results?

- Line 269: Are the results applicable to other people in the food in who may be working on other milk-based products?

Round 2

Reviewer 2 Report

Accept in present form

Author Response

Thank you for your comments

Reviewer 3 Report

Line 140: Statistical analysis was conducted on what measurements?

Author Response

Line 140: Statistical analysis was conducted on what measurements?

The statistical analysis was conducted on the milk fat experiment. The objective was to detect if changing the milk fat concentration has a siginifcant effect on the yield of Qishta. 

please check Line 287-288